# Measurement properties of the 30-second sit-to-stand test in post COVID-19 condition: Results from the PYCNOVID randomised controlled trial

**Julia Braun, Julia Kopp, Lisa Künzi, Milo A Puhan, Jan S Fehr, Thomas Radtke** *

Epidemiology, Biostatistics and Prevention Institute, University of Zurich, Zürich, Switzerland

* thomas.radtke@uzh.ch

## Abstract

### Background

The 30-second sit-to-stand test (30s-STS) is a frequently used measure of functional capacity in post-COVID-19 condition (PCC), but its measurement properties have not been comprehensively investigated.

### Methods

We used data from the PYCNOVID randomised controlled trial to examine feasibility, safety, test-retest reliability, construct validity (correlations with key symptoms and physical activity), and responsiveness to change of the 30s-STS. Data were collected at screening, at baseline (approximately 2 weeks after screening), and at follow-up (12 weeks after baseline). Using screening and baseline data, agreement was evaluated using Bland-Altman analysis (mean bias and limits of agreement), alongside the intraclass correlation coefficient (ICC) and the minimal detectable change (MDC95). Correlation coefficients were used to assess relationships between the 30s-STS test, physical activity, and key PCC-related symptoms. The minimal important difference (MID) was estimated with data from baseline and at 12-weeks using anchor- and distribution-based methods.

### Results

Data from 150 adults with PCC (74.7% female) were analysed. The 30s-STS was safe (no adverse event) and feasible. Overall, test performance was reduced, 46% of participants had z-score values below the 5th percentile. The test showed good test-retest reliability (ICC 0.82 [95% CI, 0.76 to 0.86]) with no indication for a systematic learning effect; however, the limits of agreement were wide. Mean bias between was 0.84 repetitions (95% CI, 0.26 to 1.42). The MDC95 was 4.25 repetitions. Correlations with physical activity and key PCC-related symptoms were weak. Distribution-based methods yielded MID values between 1.38 and 3.23 repetitions.

**Data availability statement:** All relevant data are within the paper and its Supporting Information files.

**Funding:** The study was funded by Horphag Research, Av. Louis-Casaï, 1216 Cointrin, Switzerland. The funder of the study had no role in study design, data collection, data analysis, data interpretation, and writing of the manuscript.

**Competing interests:** The authors have declared that no competing interests exist.

Change correlations between 30s-STS repetitions and the anchors were insufficient for anchor-based MID calculation.

## Conclusions

The 30s-STS test is feasible, safe, and demonstrates good test-retest reliability. Changes exceeding four repetitions surpass measurement error (MDC95), while distribution-based estimates suggest a provisional MID of two repetitions. Further studies are needed to establish a triangulated MID to better capture patient-relevant change.

## Trial registration

ClinicalTrials.gov, NCT05890534.

## 1. Introduction

Post-COVID-19 condition (PCC) is a multisystem disease caused by severe acute respiratory syndrome coronavirus-19 (SARS-CoV-2) infection. The prevalence of PCC in the general population is estimated at ~6–7% with major implications for healthcare systems and the economy [1]. The pathophysiology of PCC remains incompletely understood. Proposed pathomechanisms include viral persistence, immune dysregulation, inflammation, endothelial and mitochondrial dysfunction, and microbiome dysbiosis, with combinations of these processes likely contributing to disease development and progression [2,3]. The disease is characterised by a variety of persisting symptoms of which fatigue, post-exertional malaise (PEM), muscle weakness and breathing difficulties are frequent with debilitating effects on individuals' physical function and quality of life [4]. Many people with PCC have reduced exercise capacity [5–7]; underlying reasons are multifactorial and include skeletal muscle abnormalities (e.g., deconditioning, local hypoxia, electrophysical alterations), and abnormal ventilatory efficiency or low oxygen pulse [6–9]. A subgroup of individuals experiencing PEM [10], a worsening of symptoms after physical or mental exertion, exhibit skeletal muscle metabolic alterations and a shift toward fast-fatigable muscle fibres [5,9], which negatively affect their exercise capacity.

The measurement of functional capacity is recommended for people with PCC and part of a core outcome set [11]. Functional capacity is typically assessed as part of (individualised) exercise-based rehabilitation programmes, which demonstrate clinically relevant improvements in cardiopulmonary function and exercise capacity [12,13]. Several tests are available for assessment of functional exercise capacity in people with PCC, of which sit-to-stand tests have received growing attention among healthcare professionals [6,14–17]. In general, sit-to-stand tests assess an individual's ability to perform as many sit-to-stand repetitions as possible within a given time. These tests provide insights into lower limb muscle function; however, test performance also depends on several other factors including balance, coordination,

neuromuscular control, pacing strategies, symptom expectation, and motivation. Although the test may appear simple, it is a complex multi-joint neuromuscular task.

The 30-second sit-to-stand test (30s-STS) is less rigorously validated than the 1 min-STS test in people with chronic respiratory conditions [18], and measurement properties have not been comprehensively investigated in people with PCC. The test appears feasible in people with post-COVID-19 sequelae [6,14–17], is responsive to exercise interventions [14], and shows unclear relationships with symptoms such as dyspnoea (11,12). Detailed knowledge of the measurement properties of the 30s-STS is critical for the interpretation of functional capacity (screening) and for evaluating changes in functional capacity following an intervention (responsiveness). Moreover, knowledge of estimates of meaningful change is a prerequisite for future trial design and sample size calculations.

Our study was developed with input from individuals with lived experience, who raised concerns about study participation when individuals with PEMS are exposed to physical stress tests. Initially, we had planned to use the 1-min STS to assess functional capacity. However, rather than omitting functional testing altogether, we agreed on an alternative and selected the shorter 30-second version.

Recently, a minimal important difference (MID) of two repetitions has been proposed for the 30-s STS test in people with chronic obstructive pulmonary disease [19], a patient population that shares certain clinical features with individuals with PCC, including fatigue, impaired exercise capacity, and lower limb muscle weakness. However, the two populations differ in disease course, symptom variability, clinical predictability, and treatment options. In addition, chronic obstructive pulmonary disease predominantly affects older individuals, whereas PCC also affects younger and middle-aged adults, and a higher proportion of females are typically affected by PCC compared with chronic obstructive pulmonary disease. These demographic and clinical differences may limit the direct transferability of the proposed MID to individuals with PCC.

The aim of this study was to evaluate the measurement properties of the 30s-STS test for assessing functional capacity in adults with PCC. Specifically, we assessed the feasibility, test-retest reliability, construct validity, and responsiveness to change of the 30s-STS in a heterogeneous sample of individuals with PCC.

## 2. Methods

### 2.1. Study design and setting

Data were collected as part of the PYCNOVID trial [20]. This randomised controlled trial investigated the effects of a nutritional supplement with antioxidative and anti-inflammatory properties on health status (primary endpoint) in people with PCC. Notably, this parent study was not designed to improve functional capacity; it was a secondary endpoint in the main trial, which may have limited responsiveness and anchor validity.

Recruitment took place at the University of Zurich (UZH), Switzerland, between 14 June 2023 and 5 July 2024. Participants attended four study centre visits: screening, baseline (about 2 weeks later), follow-up (6 weeks after baseline), and a final visit after the 12-week intervention. Details about the study design, inclusion and exclusion criteria, and assessment methods are described in detail elsewhere [20].

The study was approved by the ethics committee of the Canton of Zurich, Switzerland (Kantonale Ethik Kommission Zürich; BASEC-Nr. 2022−01967).

### 2.2. Participants

Individuals with PCC experiencing at least one persistent PCC symptom, such as fatigue, cognitive impairment ("brain fog"), dyspnoea, or PEM [20] were eligible to participate if they were aged 18 years or older, fluent in German, had no planned changes in medication, and had a confirmed SARS-CoV-2 infection verified by polymerase chain reaction (PCR), a rapid antigen test, or a physician's diagnosis of PCC. We applied the World Health Organisation definition of PCC [21]

and assessed each participant's clinical history, including the timing of SARS-CoV-2 infection(s), vaccination(s), and the onset and progression of symptoms over time. Specifically, participants were required to have persistent symptoms for ≥12 weeks after SARS-CoV-2 infection that could not be explained by an alternative diagnosis. In the absence of a diagnostic biomarker, each case was carefully adjudicated based on medical report. Exclusion criteria comprised severe comorbidities such as renal failure or advanced heart failure, acute infections, untreated or unstable psychiatric disorders, recent COVID-19 vaccination (<4 weeks prior to baseline or during the study), intolerance to or regular use of the investigational product Pycnogenol®, or participation in another intervention study.

We recruited participants through various channels including the Altea network, the Long COVID Citizen Science Board, the Facebook group of Long COVID Schweiz, and advertisements in public transport in the city of Zurich and the surrounding metropolitan area. Furthermore, we shared flyers with different stakeholders, including clinics that care for individuals with PCC. All participants provided written informed consent prior to enrolment.

### 2.3. Data sources and measurements

Demographic characteristics and medical history were collected via online surveys and verified using medical records during the screening and baseline visits.

**2.3.1. Symptoms and patient-reported outcome measures.** Presence of PCC-related symptoms (e.g., PEM) was assessed using a 5-point Likert scale with anchors ranging from "not bad at all" to "very severe". We did not assess PEM using a validated instrument, which limits the validity of our evaluation of this complex symptom [22]. Fatigue was assessed with the 13-item FACIT-Fatigue instrument, with scores <34 indicating clinically relevant fatigue [23,24]. Dyspnoea was assessed with the Chronic Respiratory Questionnaire (CRQ) dyspnoea domain [25]. Cognitive function was assessed using the German version of the Montreal Cognitive Assessment (MoCA) test [26,27]. A cut-off score of < 26 was used to indicate impairment. Anxiety and depression were assessed with the Hospital Anxiety and Depression Scale (HADS). A cut-off value of 7 or higher was used to identify participants with depression and/or anxiety [28]. Self-reported health status was recorded daily over seven consecutive days using the EQ-VAS (0–100 scale: 0, worst imaginable health; 100, best imaginable health) [29,30], and mean values were calculated for each participant.

**2.3.2. 30-second sit-to-stand test.** The 30s-STS test was conducted three times during the study: at screening (initial trial to familiarise participants with the test procedures), at baseline and at the final visit 12 weeks post baseline. Screening and baseline tests took place within a maximum of two weeks, so that it can be assumed that no relevant changes in the participants' health status took place in this time. The test was performed on a height adjustable chair placed against the wall. The seat height was adjusted to the participants' leg length (i.e., 90° knee angle) and kept constant throughout the study to avoid measurement bias. The participants placed their hands around their hips and were instructed to perform as many repetitions as possible in 30 seconds. They were not allowed to use their hands to assist their movement or to place their hands on their legs for support. Heart rate and oxygen saturation were measured with a portable pulse oximeter (Beurer PO 80, Beurer GmbH, Ulm, Germany), and ratings of perceived exertion and dyspnoea (0–10 Borg scale) were recorded before and after the exercise test. Post-test, the lowest oxygen saturation and the highest heart rate were recorded. After each test, we asked whether the participant had exerted maximal effort or had limited their performance out of concern that overexertion might exacerbate their symptoms (herein referred to as submaximal test). Responses were recorded as yes or no.

Team members supervising the 30s-STS test underwent rigorous training prior to the start of the study. Standardised working instructions were provided to ensure consistent test administration and minimise bias. We aimed to have the same outcome assessor supervise each participant throughout the 12-week study; however, this was not always possible.

**2.3.3. Physical activity.** Physical activity was measured for eight consecutive days using ActiGraph wGT3X-BT accelerometers (Pensacola, FL, USA) recording raw acceleration at 30 Hz. Participants wore the devices on their hips during waking hours and were instructed to remove them only for water-based activities. We processed raw

accelerometer data using the GGIR package (version 3.2.6) combined with the R package read.gt3x (version 1.2.0). All analyses were performed on a local computing environment. Raw.gt3x files were auto-calibrated to local gravity (1 g ≈ 9.81 m.s$^{-2}$) and converted to 5-second epoch summaries using the Euclidean Norm Minus one (ENMO) metric, expressed in milligravity units (mg). Non-wear periods were identified and imputed according to GGIR's standard algorithms based on sustained inactivity.

Activity recordings with a minimum wear time of 10 hours per day with at least three valid weekdays and one valid weekend day were included in the analysis. All analyses were performed in parallel mode to optimise computation time, default GGIR calibration, and non-wear detection.

Data were summarised across waking hours only, as participants removed the ActiGraph during sleep. Physical activity intensities were classified using Euclidean Norm Minus One (ENMO) thresholds of 47.2 mg for light, 69.1 mg for moderate, and 258.7 mg for vigorous PA, with activity bouts defined using a 90% adherence criterion [31]. ENMO, derived from raw accelerometer data, reflects overall physical activity intensity across a full day, with higher values indicating higher activity levels.

## 2.4. Statistical analysis

All analyses were conducted with R, version 4.5.2. For descriptive statistics we report means and standard deviations for continuous data and numbers and percentages for categorical data. Z-scores for the 30s-STS repetitions were calculated from the data of the sarcopenia cohort from Copenhagen [32]. We use the 5$^{th}$ percentile (z-score −1.645) as a commonly accepted impairment threshold for descriptive purposes only; it is not intended as a diagnostic cut-off.

Missing data in this study were minimal; therefore, we did not apply imputation methods.

**2.4.1. Test-retest reliability.** Reliability refers to the consistency of a measurement and its ability to reproduce performance across repeated assessments. To assess test-retest reliability, we compared the 30s-STS results between screening and baseline using Bland-Altman plots including mean bias and limits of agreement, defined as the mean difference ±1.96 times its standard error, and calculated one-way intraclass correlation coefficients (ICC) for consistency. We opted for the one-way ICC instead of the two-way ICC because the latter assumes that all participants' measurements were supervised by the same outcome assessors, which was not the case in our study.

Measurement error comprises systematic and random deviations in a participant's performance that are not attributable to true changes in the underlying construct. We calculated the standard error of measurement (SEM), defined as the standard deviation at screening times the square root of 1 minus ICC, and the minimal detectable change (MDC95), defined as 1.96 times the square root of 2 times SEM. The calculation of these two values gives additional information about the measurement error and its influence on change. As the screening 30s-STS test served as an initial test trial, this analysis also allowed us to examine whether a familiarisation test is necessary.

In a sensitivity analysis, we excluded participants with submaximal effort to evaluate the influence on test-retest reliability indices and assess the robustness of the primary analysis.

**2.4.2. Construct validity.** Validity is a measure of how well a test measures what it sets out to measure, that is, if it relates to the reference standard measure (criterion validity) or other measures that assess the same construct.

Regarding construct validity, potential relationships between z-scores of the 30s-STS test and PEM, fatigue, dyspnoea, and physical activity were investigated using boxplots and scatterplots, along with the calculation of Spearman's correlation coefficients. We formulated *a priori* hypotheses regarding the expected direction and strength of correlations. PCC is a multisystem condition with more than 200 reported symptoms and multiple phenotypes [33,34]. In some patients, symptoms such as fatigue, PEM, or dyspnoea are likely to be associated with performance in the 30s-STS test, whereas in others this may not be the case due to substantial heterogeneity in symptom profiles and severity. Consequently, when assessed across a heterogeneous population, these variable and patient-specific associations are likely to result in overall weak correlations. In addition to PCC-related symptoms and habitual physical activity, performance on functional tests is

influenced by a range of factors, including lower limb strength, neuromuscular control, flexibility, pain, motivation, cardio-pulmonary fitness, and autonomic function.

We anticipated that higher symptom burden and lower physical activity would be associated with lower 30s-STS performance. However, given that the 30s-STS tests reflects integrative functional capacity, we expected these correlations to be weak to at most moderate, as symptoms, and physical activity are not direct measures of functional capacity or lower limb muscle function. Furthermore, we hypothesised that subgroups of people with severe-to-very severe PEM, which represents a significant barrier to sustained (structured) physical activity, would demonstrate the lowest 30s-STS performance.

As closely related objective measures of lower-limb muscle function were not available, convergent validity could only be assessed indirectly. Therefore, the examined variables were expected to show weak correlations, reflecting related but distinct constructs.

**2.4.3. Minimal important difference.** Responsiveness is the ability of a measurement instrument to detect change over time in the construct of interest, including clinically meaningful changes, even when those changes are small. To determine the MID for the 30s-STS test, we used a combination of anchor-based and distribution-based approaches [35–38]. Specifically, we used four different distribution-based approaches to calculate MIDs: 1) standard deviation based approach (0.5*SD baseline); 2) standard error of measurement based approach (SD baseline*square root[1-ICC]); 3) Cohen's effect size (0.5*SDΔ), and 4) empirical rule effect size (0.08*6*SDΔ) [35–38]. With respect to the anchor-based approach, we examined the change from baseline to 12 weeks in the 30s-STS test and anchor variables, where some change was expected. Several potential anchors were selected: CRQ dyspnoea [39], FACIT Fatigue score [40], HADS total score [41,42] along with the two subscores for depression and anxiety as well as the EQ-VAS [29]. Boxplots and Q-Q-plots were used to check for normal distribution and the number of changers, i.e., the number of measurement differences that exceeded the known MIDs of the respective anchors, was calculated. To calculate MIDs, we used linear regression and receiver operating characteristic (ROC)-based methods. For linear regression-based methods scatter plots had to show a linear relationship and the absolute value of Spearman's correlation coefficients between the change of 30s-STS repetitions and the change of the respective anchor was pre-defined to exceed 0.3 [43]. For the ROC-based method, the area under the ROC curve (AUC) was calculated to assess how well the change in 30s-STS repetitions predicted improvements in the anchors exceeding their respective MIDs. Only AUC values above 0.7 were deemed acceptable for further analysis.

**2.4.4. Reporting and methodological considerations.** This is a secondary analysis of data from a placebo-controlled randomised controlled trial [20]; therefore, we did not perform an *a priori* sample size calculation. Moreover, the analysis does not focus on between-group comparisons. We therefore follow the Strengthening the Reporting of Observational Studies in Epidemiology (STROBE) guidelines for reporting observational studies [44].

The methodology for assessing reliability, validity, reliability, and responsiveness of the 30s-STS test were guided by COSMIN recommendations (Version 2.0) [45], with adaptations made to account for the fact that the 30s-STS is a functional capacity test rather than a patient-reported outcome measure.

## 3. Results

One-hundred fifty-three participants were randomised at the baseline visit and were included in the primary study. The 30s-STS test was available for 152 participants at screening (99.3%), 150 participants at baseline (98.0%), and 144 participants at study end (94.1%). Table 1 and S1 Table present characteristics of all participants who performed a 30s-STS at baseline. Most participants were female (74.7%), and the mean age was 45 years (range 18–80 years). The percentage of participants with fatigue was high (82.7% had a FACIT Fatigue score <34), and mean EQ-VAS values were remarkably low, indicating poor perceived health status.

**Table 1. Participants characteristics at baseline.**

| Characteristics | All (n = 150) |
|---|---|
| Female sex, n (%) | 112 (74.7) |
| Age, years | 44.6 ± 12.6 |
| Hospitalized for SARS-CoV-2 infection, n (%) | 6 (4.0) |
| Time since onset of symptoms, weeks | 111 ± 58.1 |
| **Comorbidities** | |
| Cardiovascular Disease, n (%) | 7 (4.7) |
| Hypertension, n (%) | 11 (7.3) |
| Diabetes, n (%) | 3 (2.0) |
| Obesity, n (%) | 28 (18.7) |
| Chronic Respiratory Disease, n (%) | 21 (14) |
| Chronic Kidney Disease, n (%) | 3 (2.0) |
| Autoimmune Disease, n (%) | 7 (4.7) |
| Psychiatric Disease, n (%) | 20 (13.3) |
| **Patient-reported outcomes*** | |
| FACIT-Fatigue | 23.6 ± 9.2 |
| MoCA | 27.6 ± 1.8 |
| CRQ – Fatigue | 3.5 ± 1.0 |
| CRQ – Dyspnoea | 5.7 ± 1.2 |
| CRQ – Emotional | 4.6 ± 1.0 |
| CRQ – Mastery | 5.7 ± 0.9 |
| HADS – Anxiety | 5.96 ± 3.80 |
| HADS – Depression | 6.56 ± 3.79 |
| EQ-5D-5L index | 0.60 ± 0.23 |
| EQ-VAS | 50.4 ± 19.4 |
| **Physical activity (n = 141)** | |
| ENMO, mg | 9.34 ± 3.65 |
| Inactivity, min.day$^{-1}$ | 1050 ± 98 |
| Light PA, min.day$^{-1}$ | 26 ± 9 |
| Moderate PA, min.day$^{-1}$ | 48 ± 27 |
| Vigorous PA, min.day$^{-1}$ | 0.8 ± 1.8 |

Data are presented as number (percentages) or mean ± standard deviation (SD). CRQ, Chronic Respiratory Questionnaire; ENMO, Euclidean Norm Minus One; EQ-5D-5L; EuroQol 5-Dimension 5-Level, EQ-VAS; EuroQol Visual Analogue Scale (0–100); HADS, Hospital, Anxiety and Depression Scale; MoCA, Montreal Cognitive Assessment Test; PA, physical activity. *Data for the Chronic Respiratory Questionnaire and EuroQol 5-Dimension 5-Level EQ-VAS were available for 149 participants.

## 3.1. Feasibility

In total, 11 participants did not perform the 30s-STS test during at least one of the three study visits, due to reasons such as feeling unwell, dizziness, recent knee surgery, or muscle pain. No adverse events were recorded during exercise testing. Table 2 summarises results for the 30s-STS test at screening and baseline visits. Overall, functional capacity was reduced, with mean (SD) z-scores for the 30s-STS test of −1.6 (1.1) and −1.4 (1.2), respectively (Table 2). Overall, 30s-STS performance varied widely; yet at baseline, 69/150 (46%) had z-scores < −1.645 (5th percentile) suggesting impaired functional capacity (Fig 1). The test elicited mild dyspnoea and leg fatigue, while oxygen saturation remained stable

**Table 2. Functional capacity (30s sit-to-stand) data at screening and baseline.**

| | Screening (n = 152) | Baseline (n = 150) | Difference (n = 150) |
|---|---|---|---|
| Repetitions | 14.8 ± 5.8 | 15.7 ± 6.5 | 0.8 ± 3.6 |
| Repetitions z-score | −1.6 ± 1.1 | −1.4 ± 1.2 | 0.1 ± 0.6 |
| Heart rate – pre-test beats.min⁻¹ | 71 ± 11 | 74 ± 11 | 2.7 ± 10.2 |
| Heart rate – post-test beats.min⁻¹ | 98 ± 18 | 100 ± 17 | 1.7 ± 14.8 |
| SpO₂ – pre-test % | 98 ± 1 | 98 ± 1 | −0.2 ± 1.5 |
| SpO₂ – post-test % | 98 ± 1 | 98 ± 2 | −0.3 ± 1.5 |
| Dyspnoea – pre-test | 0.8 ± 1.1 | 0.7 ± 1.0 | −0.1 ± 1.0 |
| Dyspnoea – post-test | 2.9 ± 1.8 | 2.7 ± 1.5 | −0.2 ± 1.4 |
| Leg fatigue – pre-test | 1.7 ± 1.8 | 1.8 ± 1.5 | 0.2 ± 1.6 |
| Leg fatigue – post-test | 3.8 ± 2.1 | 3.6 ± 1.8 | −0.1 ± 1.6 |

Data are presented as mean ± standard deviation. SpO₂, oxygen saturation. Dyspnoea and leg fatigue were assessed on a 0–10 Borg scale.

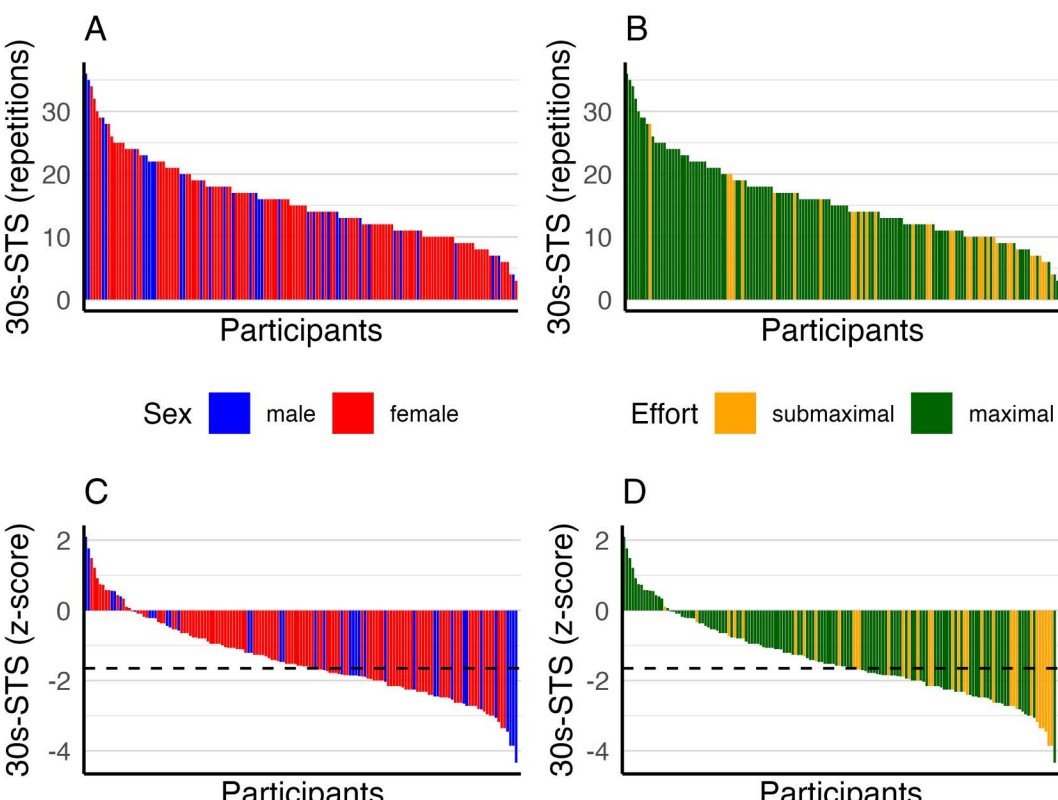

**Fig 1. Barplot of individual 30s-STS repetitions (A, B) and resulting z-scores (C, D) by sex and effort.** The dashed black line in panels C and D represents the 5th percentile, corresponding to a z-score of −1.645.

post-test. The mean differences for 30s-STS repetitions and related variables between screening and baseline were small and without clinical relevance (Table 2).

Thirty-four of the 150 participants (22.7%) reported limiting their effort out of fear that symptoms might worsen, suggesting submaximal performance. Those with submaximal effort had lower 30s-STS z-scores compared to those who reported maximum effort: mean (SD) z-scores −1.23 (1.14) versus −2.11 (0.97), see Table 3 and Fig 1. There were no relevant differences in 30s-STS related variables (Table 3) or demographic and clinical characteristics including presence of comorbidities, symptom burden, perceived health status and quality of life between those with maximal versus submaximal effort (Table S2).

### 3.2. Test-retest reliability

The mean difference of 30s-STS repetitions between screening and baseline was 0.8 repetitions (Table 2). The Bland-Altman plot (Fig 2) reveals no apparent bias, but the limits of agreement were wide. Mean bias was 0.84 repetitions (95%

**Table 3. Functional capacity (30s sit-to-stand) data at baseline, stratified by effort.**

| | Maximal effort (n = 116) | Submaximal effort (n = 34) |
|---|---|---|
| Repetitions, n | 16.8 ± 6.5 | 12.2 ± 5.1 |
| Repetitions, z-score | −1.2 ± 1.1 | −2.1 ± 1.0 |
| Heart rate – pre-test, beats.min$^{-1}$ | 73 ± 11 | 76 ± 13 |
| Heart rate – post-test, beats.min$^{-1}$ | 101 ± 17 | 97 ± 19 |
| SpO$_2$ – pre-test, % | 98 ± 1 | 98 ± 1 |
| SpO$_2$ – post-test, % | 98 ± 2 | 98 ± 1 |
| Dyspnoea – pre-test, 0–10 scale | 0.7 ± 1.0 | 0.7 ± 1.1 |
| Dyspnoea – post-test, 0–10 scale | 2.8 ± 1.5 | 2.4 ± 1.4 |
| Leg fatigue – pre-test, 0–10 scale | 1.8 ± 1.4 | 1.9 ± 1.8 |
| Leg fatigue – post-test, 0–10 scale | 3.7 ± 1.7 | 3.4 ± 1.9 |

Data are presented as mean ± standard deviation. SpO$_2$, oxygen saturation. Dyspnoea and leg fatigue were assessed on a 0–10 Borg scale.

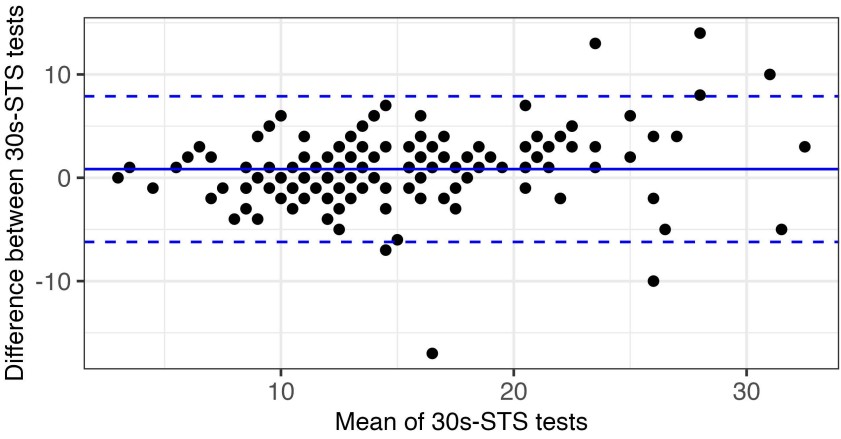

**Fig 2. Bland-Altman plot for the agreement between 30s-STS repetitions at screening and at baseline.** The blue line represents the mean bias of 0.84 (95% confidence interval from 0.26 to 1.42). The dashed lines represent the lower and upper limit of agreement, with respective values of −6.21 (95% confidence interval from −7.20 to −5.22) and 7.89 (95% confidence interval from 6.89 to 8.88).

CI, 0.26 to 1.42) with a lower limit of agreement of −6.21 repetitions (95% CI, −7.20 to −5.22) and upper limit of agreement of 7.89 repetitions (95% CI, 6.89 to 8.88). Altogether, 7/150 participants had values outside the limits of agreement. The one-way ICC was 0.82 (95% CI, 0.76 to 0.86) indicating good reliability and no indication for a systematic learning effect. The SEM was 2.35 repetitions, with a corresponding MDC95 of 4.25 repetitions, indicating that changes greater than four repetitions can be interpreted as exceeding measurement error.

A sensitivity analysis excluding participants with submaximal effort yielded a slighter higher mean bias of 1.17 repetitions (95% CI, 0.48 to 1.87), but limits of agreement remained relatively comparable (S1 Fig).

### 3.3. Construct validity

Fig 3 shows relationships between FACIT Fatigue, CRQ fatigue, CRQ dyspnoea, PEM and the 30s-STS repetitions at baseline. Overall, correlation coefficients ranged between 0.22 and 0.32, denoting weak correlations. Consistent with our hypothesis, boxplots of STS performance across the four PEM grades suggested that individuals with very severe PEM completed fewer 30s-STS repetitions, whereas the other three PEM groups displayed comparable values. Similarly, relationships between physical activity and 30s-STS z-scores were weak with correlation coefficients between 0.22 and 0.26, see S2 Fig.

### 3.4. Responsiveness to change

All potential anchors used in the triangulation process to calculate MID values for 30-s STS repetitions were visually inspected using boxplots and Q–Q plots (S3 Fig). Based on this inspection, the distributions appeared at least approximately normal. They showed a sufficiently large number of participants with positive change, with percentages ranging from 25% to 50%; therefore, all anchors qualified for use in the MID calculations. Table 4 displays Spearman's correlation coefficients and AUC values of the change in the anchors and the change in 30s-STS repetitions. Correlation coefficients

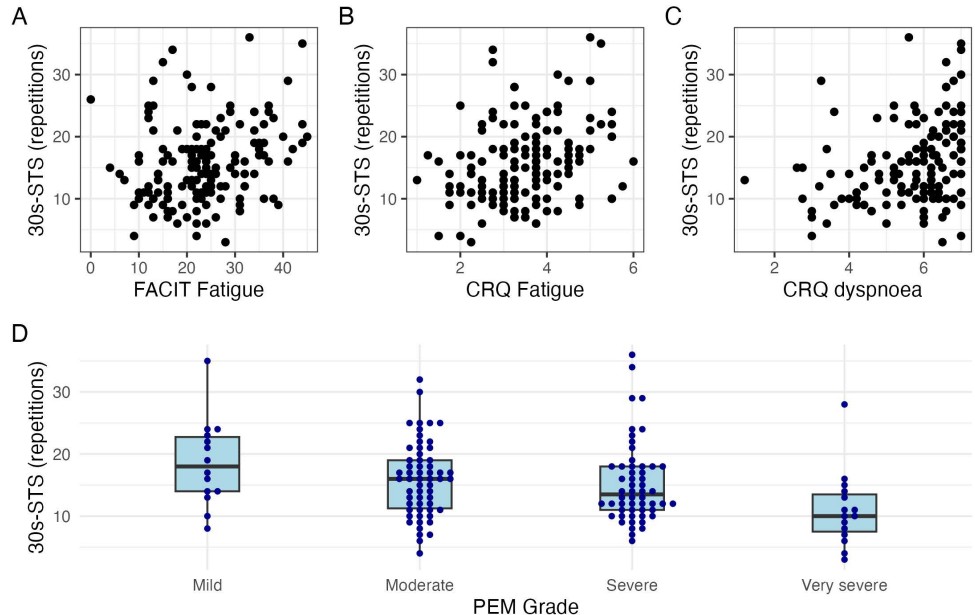

**Fig 3. Scatterplots showing relationships at baseline between 30s-STS repetitions and FACIT Fatigue (A), Chronic Respiratory Questionnaire (CRQ) fatigue domain (B), and Chronic Respiratory Questionnaire (CRQ) dyspnoea domain (C).** Boxplots depicting relationships at baseline between post-exertional malaise (PEM severity grades assessed on a 5-point Likert scale) and 30s-STS repetitions **(D)**. Spearman correlation coefficients were as follows: (A) r = 0.22; (B) r = 0.29; (C) r = 0.32.

**Table 4. Correlation coefficients and area under the curve (AUC) values for the relationship between the change in 30s-STS repetitions and potential anchor variables from baseline to 12 weeks.**

|  | Correlation coefficient | AUC |
|---|---|---|
| CRQ Dyspnoea | 0.24 | 0.63 |
| FACIT Fatigue | 0.13 | 0.56 |
| HADS total | −0.20 | 0.61 |
| HADS anxiety | −0.19 | 0.60 |
| HADS depression | −0.14 | 0.52 |
| EQ-VAS | 0.21 | 0.60 |

Data are Spearman correlation coefficients. AUC, area under the curve; CRQ, Chronic Respiratory Questionnaire; EQ-VAS, EuroQol Visual Analogue Scale. HADS, Hospital Anxiety and Depression Scale.

were all below 0.3 indicating weak correlations; none of the anchors seemed appropriate for use in an MID calculation. Similarly, using the ROC-based method, all AUC values were below the threshold of 0.7; therefore, no MID values were calculated. Further details are provided in the online supplementary material (S3 and S4 Tables, S3 and S4 Figs).

The four distribution-based approaches yielded MID estimates between 1.38 and 3.23 repetitions for the 30s-STS test (Table 5). We propose a provisional triangulated MID of 2 repetitions for the 30s-STS in people with PCC.

## 4. Discussion

This study evaluated the measurement properties of the 30s-STS test in a large and heterogeneous sample of adults with PCC and high symptom burden. Following key findings emerged. First, the 30s-STS test was feasible and safe to administer, demonstrated good test-retest reliability, and showed no evidence for a learning effect. However, measurement error and between-subject variability can influence test results and should be considered when interpreting 30s-STS results. Second, performance on the 30s-STS test was reduced in people with PCC, suggesting impaired integrated functional capacity. Third, relationships between the 30s-STS test, habitual physical activity and key PCC-related symptoms were weak, consistent with our hypothesis. Fourth, estimation of a MID using anchor-based approaches was not feasible due to weak correlations between changes in 30s-STS performance and the anchor measures. Distribution-based approaches suggest a MID of approximately two repetitions for the 30s-STS test; however, this preliminary estimate requires prospective evaluation.

**Table 5. Minimal important difference values for 30s-STS test repetitions calculated using different distribution-based methods.**

| Distribution-based approach | Calculation | MID (repetitions) |
|---|---|---|
| Standard deviation | 0.5*SD baseline | 3.23 |
| Standard error of measurement | SD baseline*sqrt[1-ICC] | 1.38 |
| Cohen's effect size | 0.5*SDD | 1.50 |
| Empirical rule effect size | 0.08*6*SDD | 1.44 |
| **Proposed MID** | **–** | **2** |

ICC, Intraclass correlation coefficient; MID, minimal important difference; SD, standard deviation; SDD, change in standard deviation from baseline to follow-up (12 weeks); SE, standard error; sqrt, square root. For the standard error of measurement approach, we calculated intraclass correlation coefficients from a random-effects model using 30s-STS tests from screening and baseline visits (~2 weeks apart).

Consistent with previous reports [6,14,16], the 30s-STS can be safely administered to assess functional capacity in people with PCC. We did not observe a learning effect when the test was repeated within approximately two weeks. The mean bias was 0.84 repetitions (95% CI, 0.26 to 1.42) indicating no clinically relevant difference at the study population level. However, given the large interindividual-variability (wide limits of agreement and a measurement error of ± 2.35 repetitions) in a clinical population with fluctuation symptoms, a familiarisation test seems meaningful.

In our study sample, functional capacity was reduced, with participants performing on average 15−16 repetitions on the 30s-STS and showing corresponding z-scores of approximately −1.4 to −1.6. While we noted large heterogeneity in 30s-STS performance with values ranging from 3 to 36 repetitions, about 46% had z-scores < −1.645, suggesting impaired functional capacity [46,47]. Although comparisons with other studies are challenging due to differences in study populations, sampling methods, inclusion and exclusion criteria, and administration of the 30s-STS test, our values fall within the range previously reported by others (11−19 repetitions) [6,14,16]. Importantly, almost one quarter reported limiting their effort out of fear that symptoms might exacerbate, influencing their 30s-STS results. This group had remarkably lower 30s-STS z-scores compared to those who reported maximum effort (−2.11 versus −1.23). We did not identify any characteristics that explain these findings, such as more severe symptom burden, lower physical activity levels, or presence of chronic disease(s) other than PCC. The two groups were well comparable with respect to demographic and clinical characteristics and had reported similar quality of life (S2 Table). Notably, this aspect is important to consider because some people with PCC may be capable of performing better in the 30s-STS but chose not to due to fear of symptom exacerbation ("crashes"). Consequently, submaximal effort may lead to an underestimation of individual functional performance, with implications for the validity of the test results at the individual level and their clinical interpretation. From a clinical perspective, it is therefore important to assess fear of symptom exacerbation as part of exercise testing in people with PCC. While this group may present a distinct clinical phenotype, our data do not permit formal identification or detailed characterisation of such subgroups. Further research is needed to explore these potential phenotypes and their impact on functional assessment and clinical outcomes.

Our participants reported high burden of PCC-related symptoms, in particular fatigue (82.7%) and PEM (94%). Relationships between the 30s-STS test and frequent PCC-related symptoms were weak, suggesting that symptom severity is not necessarily a limiting factor for a short and specific functional task such as the 30s-STS test. Several factors may contribute to this decoupling. First, participants may adopt pacing or anticipatory behaviours, consciously or subconsciously limiting effort to avoid symptom exacerbation, which can mask the expected relationship between symptoms and performance. Second, although some individuals may fear symptom worsening, it is unclear whether such a brief, 30-second task is sufficient to trigger PEM. Finally, there may be a temporal mismatch between the symptom measurement windows and the acute performance during the test, such that momentary functional capacity is not fully captured by symptom questionnaires.

The results of previous studies investigating relationships between 30s-STS performance and PCC-related symptoms are heterogeneous [6,16,17]. Our data are consistent with a previous study of 102 individuals with PCC showing no relationships between the 30s-STS test and fatigue, PEM, myalgia and dyspnoea [16]. We studied a heterogenous group of people with PCC, and between-subject variability in those reporting fatigue (herein defined as FACIT-Fatigue scores <34) was wide, ranging from −4–2 z-score units. With respect to PEM, as expected, the group of participants reporting very severe PEM had lower 30s-STS z-scores compared to those reporting either mild, moderate or severe PEM. However, these findings must be interpreted with caution since we did not use a validated instrument to assess PEM [22] but rather a single question with a Likert-type severity grading, which has limited validity. Taken together, these considerations suggest that short-duration functional tests provide complementary information to symptom assessments but may not fully reflect the impact of PCC-related symptoms on daily life.

We further aimed to establish a triangulated MID for the 30s-STS test by using a combination of distribution-based and anchor-based approaches. With respect to the anchor-based approaches, the strength of correlation and AUC

 

values between the change in 30s-STS test repetitions and change in the different anchors from baseline to 12-weeks was insufficient for an MID calculation (Table 4). This is supported by the overall weak correlations between the 30s-STS test and key PCC-related symptoms (construct validity), suggesting that these symptoms do not adequately reflect the construct of lower limb muscle weakness. The placebo-controlled randomised controlled trial investigating effects of Pycnogenol®, a nutritional supplement with antioxidative and anti-inflammatory properties, from which the data for this analysis were obtained [20], did not produce sufficient variation or change to allow for anchor-based MID calculations. The four distribution-based approaches yielded MID values between 1.38 and 3.23 repetitions in the 30s-STS test. Until an anchor-based MID becomes available, we propose an MID of at least two repetitions for 30s-STS test results to be used in sample size calculations in studies focusing on functional capacity in people with PCC. This estimate is identical to the MID proposed for people with chronic obstructive pulmonary disease [19], a patient population that shares several characteristics with individuals with PCC, including breathing difficulties, fatigue, impaired exercise capacity, and lower limb muscle weakness. Moreover, recent data from a meta-analysis of five exercise-based studies using the 30s-STS test demonstrated an increase of 3.05 repetitions (95% CI 1.96–4.13) compared with control (no training), suggesting that our MID estimate falls within the range of observed values considered clinically meaningful [12].

Importantly, as the MDC95 of 4.25 repetitions suggests that changes below this threshold might be primarily or partly due to measurement error and between-subject variability, the proposed MID value of two repetitions must be interpreted with caution. A change of two repetitions could fall within measurement error, while changes above 4 repetitions can be interpreted as both real and meaningful. However, using a comparatively small MID of two repetitions for sample size calculations represents a conservative approach in the absence of further evidence: a smaller MID results in a larger sample size, thereby ensuring sufficient power for future studies.

Among the limited treatment options for PCC, exercise-based rehabilitation has been shown to reduce symptom burden and produce clinically meaningful improvements in exercise capacity [12,13]. Such interventions would therefore be well suited for determining MIDs using anchor-based approaches. We propose to use the six-minute walk test (6MWT) and lower limb muscle function measurements (e.g., dynamometry) as suitable anchors, in addition to global impression of change question(s) to incorporate the patient perspective [48]. For example, the 6MWT shows moderate to strong correlations with the 30s-STS test in healthy people and those with heart failure and chronic obstructive pulmonary disease (r = 0.61 to 0.65) [19,49,50]; the test has a well-established MID of 25-33m [51]; and current evidence from a systematic review and meta-analysis shows clinically meaningful improvements in 6MWT distance after exercise-based rehabilitation compared to usual care (mean difference 89.5 m; 95% CI 9.9–169.2) [12]. This suggests the potential of the 6MWT as a valuable anchor for MID calculations.

## 4.1. Strengths and limitations

Strengths of this study include the large and heterogeneous sample, which covers a wide age range and a broad spectrum of PCC severity, suggesting that our findings are applicable to a broad adult PCC population. People with lived experience were actively involved in discussions around study design, outcome selection, and the choice of exercise test. Limitations include lack of reference values for the 30s-STS from a Swiss population. We therefore used data from a Danish Cohort (Copenhagen Sarcopenia Cohort) [32], a cohort of persons aged 20–93 years to calculate z-scores for test interpretation. We do not believe that the reference dataset introduced systematic error (e.g., under- or overestimation of 30s-STS performance at the population level). Both Switzerland and Denmark are high-income European countries, and the proportion of physical inactivity (i.e., not meeting the World Health Organisation physical activity guidelines) appears to be comparable: Denmark: 28.5% (95% CI, 22.7–35.0) and Switzerland: 29.3% (95% CI, 21.2–38.8) [52]. Given that physical activity and exercise capacity are typically correlated [53], similar levels of physical inactivity support the assumption that these countries have broadly comparable functional capacity profiles. Importantly, use of the Danish reference data and corresponding equations should be considered an approximate benchmark rather than a definitive classification of

impairment. Moreover, we did not assess functional capacity with another test, such as the 6MWT or a measure of lower limb muscle function, because this was not the focus of the primary study [20]. Such data would have been valuable for construct validity and as anchors for calculating the MID. Consequently, the proposed MID of at least two repetitions relies solely on distribution-based (statistical) methods and does not capture the patient perspective.

## 5. Conclusions

The 30-second sit-to-stand test is feasible, safe, and demonstrates good test-retest reliability. Distribution-based methods suggest a provisional minimal important difference of at least two repetitions, whereas the minimal detectable change is about four repetitions. Future studies are warranted to establish a triangulated minimal important difference using a combination of anchor- and distribution-based approaches to capture the patient perspective.

## Supporting information

**S1 Table. Participant characteristics at baseline.**
(DOCX)

**S2 Table. Descriptive characteristics of the groups with maximal and submaximal effort in the 30-second sit-to-stand test (30s-STS).**
(DOCX)

**S3 Table. 30-second sit-to-stand data at 12 weeks and change from baseline to 12 weeks.**
(DOCX)

**S4 Table. Minimal important difference (MID) values used for the potential anchors and associated number of changers (n = 144).**
(DOCX)

**S1 Fig. Bland-Altman plot for the agreement between 30-second sit-to-stand (30s-STS) repetitions at screening and at baseline, excluding participants with submaximal effort.** The blue line represents the mean bias of 1.17 (95% confidence interval from 0.48 to 1.87). The dashed lines represent the lower and upper limit of agreement, with respective values of −6.26 (95% confidence interval from −7.46 to −5.07) and 8.61 (95% confidence interval from 7.41 to 9.80).
(DOCX)

**S2 Fig. Scatterplots showing relationships at baseline between 30-second sit-to-stand test (30s-STS) z-scores and daily steps (A), Euclidean Norm Minus One (ENMO) values (B), time spent in daily moderate physical activity (C), and time spent in daily vigorous physical activity (D).** Spearman correlation coefficients were as follows: r = 0.26 (A), r = 0.22 (B), r = 0.20 (C), and r = 0.22 (D).
(DOCX)

**S3 Fig. A) Boxplots and B) Q-Q plots for the assessment of the distribution of change from baseline to 12 weeks of the anchors and 30s-STS repetitions.**
(DOCX)

**S4 Fig. Scatterplots of the relationship between the change of the anchor variables and the 30s-STS repetitions between baseline and 12 weeks.**
(DOCX)

**S1 File. Raw data.**
(XLSX)

**S2 File. STROBE_Checklist_PLOSONE.**
(DOCX)

## Acknowledgments

We kindly thank the whole study team for their valuable work and commitment to the study. Moreover, we thank Dara Corina Spies Rodriguez for her support with data monitoring, data cleaning, and data preparation. We also thank the people with lived experience from the Long COVID Citizen Science Board for their input on study design and all institutions and healthcare professionals who supported us with participant recruitment. Most importantly, we thank all study participants for their time, effort, and willingness to contribute to this research.

## Author contributions

**Conceptualization:** Julia Kopp, Lisa Künzi, Milo A Puhan, Jan S Fehr, Thomas Radtke.

**Data curation:** Julia Braun, Julia Kopp.

**Formal analysis:** Julia Braun.

**Funding acquisition:** Milo A Puhan, Jan S Fehr, Thomas Radtke.

**Investigation:** Julia Kopp, Lisa Künzi, Jan S Fehr, Thomas Radtke.

**Methodology:** Julia Braun, Julia Kopp, Lisa Künzi, Milo A Puhan, Jan S Fehr, Thomas Radtke.

**Project administration:** Julia Kopp, Lisa Künzi, Thomas Radtke.

**Resources:** Julia Kopp, Lisa Künzi, Milo A Puhan, Thomas Radtke.

**Supervision:** Milo A Puhan, Jan S Fehr, Thomas Radtke.

**Validation:** Julia Braun, Milo A Puhan, Thomas Radtke.

**Visualization:** Julia Braun.

**Writing – original draft:** Thomas Radtke.

**Writing – review & editing:** Julia Braun, Julia Kopp, Lisa Künzi, Milo A Puhan, Jan S Fehr.

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
