## [Decision Letter · Decision Letter 0]

3 Mar 2026

PONE-D-26-04488Measurement properties of the 30-second sit-to-stand test in post COVID-19 condition: results from the PYCNOVID randomised controlled trialPLOS One

Dear Dr. Radtke,

Thank you for submitting your manuscript to PLOS ONE. After careful consideration, we feel that it has merit but does not fully meet PLOS ONE’s publication criteria as it currently stands. Therefore, we invite you to submit a revised version of the manuscript that addresses the points raised during the review process. Please submit your revised manuscript by Apr 17 2026 11:59PM. If you will need more time than this to complete your revisions, please reply to this message or contact the journal office at plosone@plos.org. . Please include the following items when submitting your revised manuscript:

If applicable, we recommend that you deposit your laboratory protocols in protocols.io to enhance the reproducibility of your results. Protocols.io assigns your protocol its own identifier (DOI) so that it can be cited independently in the future. For instructions see: https://journals.plos.org/plosone/s/submission-guidelines#loc-laboratory-protocols. Additionally, PLOS ONE offers an option for publishing peer-reviewed Lab Protocol articles, which describe protocols hosted on protocols.io. Read more information on sharing protocols at . Additionally, PLOS ONE offers an option for publishing peer-reviewed Lab Protocol articles, which describe protocols hosted on protocols.io. Read more information on sharing protocols at https://plos.org/protocols?utm_medium=editorial-email&utm_source=authorletters&utm_campaign=protocols..

We look forward to receiving your revised manuscript.

Kind regards,

Mukhtiar Baig, Ph.D.

Academic Editor

PLOS One

Journal Requirements:

Reviewers' comments:

Reviewer's Responses to Questions

**Comments to the Author**

1. Is the manuscript technically sound, and do the data support the conclusions?

Reviewer #1: Yes

Reviewer #2: No

Reviewer #3: Partly

2. Has the statistical analysis been performed appropriately and rigorously? 

Reviewer #1: Yes

Reviewer #2: No

Reviewer #3: Yes

3. Have the authors made all data underlying the findings in their manuscript fully available?

Reviewer #1: Yes

Reviewer #2: Yes

Reviewer #3: Yes

4. Is the manuscript presented in an intelligible fashion and written in standard English?

Reviewer #1: Yes

Reviewer #2: Yes

Reviewer #3: Yes

5. Review Comments to the Author

Reviewer #1: This manuscript examines the 30-second sit-to-stand (30s-STS) test in individuals with post-COVID-19 condition (PCC). Overall, the study is well conducted, clearly reported, and methodologically sound.

However,

1) The manuscript reports weak correlations between 30s-STS performance and key PCC-related symptoms, stating that this was “expected.” However, the theoretical basis for this expectation is not sufficiently articulated. The authors should more clearly define and explain why weak correlations are theoretically plausible.

2) Considering 22.7% of participants reported submaximal effort due to fear of symptom exacerbation. This is a clinically important issue in PCC. While a sensitivity analysis is reported, the authors should explicitly discuss how this behaviour affects the validity and clinical interpretation of the 30s-STS.

3) The images in the figures are blurry; please upload sharp versions of these images.

Reviewer #2: Please describe the measurement properties you evaluated in the methodology section. And explain in detail how you evaluated them. You haven't provided any information about these. Also, which methodological criteria did you use? For example, the most commonly used guideline currently is the COSMIN checklist. Did you base your analysis on this? For instance, COSMIN does not recommend analyzing psychometric properties using data from previously conducted longitudinal studies. For detailed information, please refer to the guideline at cosmin.nl.

At what time intervals did you examine the test-retest reliability? Before and 12 weeks later? If so, that's a very significant time gap and not appropriate.

You mentioned in the abstract that you looked at construct validity, but the methodology doesn't specify what you compared it to. What hypotheses do you have for divergent validity or convergent validity? Example article: Tolk et al. Measurement properties of the OARSI core set of performance-based measures for hip osteoarthritis: a prospective cohort study on reliability, construct validity and responsiveness in 90 hip osteoarthritis patients.

Line 150: "...to assess a potential learning effect"... You are looking at reliability, and please remove this statement. Clearly specify the analysis sections for reliability or other measurement characteristics.

I don't mean to be disrespectful, but I think you're not generally familiar with this type of study design. You haven't explained the measurement characteristics, the analysis isn't appropriate, and you haven't followed a methodology. Nowadays, these kinds of validity and reliability studies undergo a rigorous methodology, and there are many examples of this in the literature.

Reviewer #3: Dear Authors,

First, I would like to commend you for conducting this important and timely study. The manuscript presents a clearly structured investigation evaluating the measurement properties of the 30-second sit-to-stand (30s-STS) test in individuals with post-COVID-19 condition (PCC). The topic is clinically relevant and methodologically meaningful, particularly given the need for valid and responsive outcome measures in PCC rehabilitation.

While the study provides valuable insights, several areas require further clarification and scientific strengthening to meet the standards of a Q1-ranked journal. My detailed comments are outlined below.

Abstract

The background is appropriately introduced; however, the study aim should be stated more explicitly and concisely.

Lines 25–30: Please provide more detailed information about the study sample (e.g., recruitment source, key inclusion criteria, baseline severity).

You report “Mean bias 0.84 repetitions” and ICC (p1, l33–35). Consider including a clinically interpretable reliability metric (e.g., SEM or MDC95), as ICC alone is dependent on between-subject variability and may overestimate practical reliability.

Since anchor-based methods were unsuccessful, the term “minimal important difference” should be used more cautiously. It would be more precise to describe this as a “distribution-based estimate of minimal detectable/important change,” unless a clear distinction between MID and MDC is consistently maintained throughout the manuscript.

Introduction

The rationale for assessing functional capacity in rehabilitation and core outcome sets is well presented. However, the current gap regarding 30s-STS measurement properties in PCC is not sufficiently developed. For a Q1-level manuscript, three short paragraphs are insufficient to fully justify novelty and contribution. The Introduction should more convincingly articulate why this investigation meaningfully advances the field.

Specific suggestions:

Line 67: The transition to the STS test is appropriate; however, I strongly recommend briefly describing the biomechanical and physiological characteristics of the STS movement. Even though prior studies have focused on single-leg STS, biomechanical insights from those works (e.g., DOI: 10.1519/JSC.0000000000004489; DOI: 10.1080/1091367X.2024.2377096) may help frame STS as a multi-joint, power-demanding, neuromuscular task rather than merely a repetition count.

At several points, the text implies that 30s-STS reflects muscle weakness. However, STS performance is influenced by multiple factors including balance, coordination, pacing strategy, symptom anticipation, motivation, and psychological factors. It would strengthen conceptual clarity to explicitly define the construct the 30s-STS is intended to represent in PCC (e.g., lower-limb functional endurance, integrated functional capacity).

You mention that the 30s-STS is less rigorously validated than the 1-min STS in chronic respiratory disease (p2, l69–70). Please add a short rationale for selecting the 30s protocol (e.g., symptom tolerance, PEM concerns, clinical feasibility).

Clarify why MID estimation is particularly critical in PCC research and why COPD-derived MIDs cannot be assumed transferable without validation.

Methods

The Methods section is detailed and generally well organized. However, several important refinements are required.

Since this is a supplement trial, expected changes may be small. This directly impacts anchor-based responsiveness analysis. Please clarify early in Methods that the parent intervention was not primarily designed to improve functional capacity, which may constrain responsiveness and anchor validity.

PCC diagnosis requires clearer operational definition. While “physician’s diagnosis” is mentioned (p4, l95–96), alignment with WHO criteria (symptom duration, exclusion of alternative diagnoses) should be explicitly stated for international readership.

PEM is assessed via a Likert question (p4, l101–103). For Q1-level rigor, justify this choice within Methods and acknowledge measurement limitations earlier rather than only in Discussion.

Screening is described as a “familiarisation test” (p4, l113–114), yet conclusions later state that familiarisation is not necessary. Consider clarifying that screening served as an initial trial, and analyses were conducted to determine whether a separate familiarisation session was required.

Section 2.3 contains extensive information spanning multiple constructs. Consider subdividing into clearer subsections (e.g., Symptom Measures, Functional Testing, Physical Activity Assessment) to improve readability and conceptual clarity.

The definition of submaximal effort (“fear of symptom exacerbation,” p5, l123–125) is clinically important. Please clarify how this variable was coded (binary, ordinal?) and whether standardized instructions were used to reduce reporting bias.

The statistical section is comprehensive and well structured. However, the ICC model choice requires stronger justification. You report a one-way ICC for consistency (p6, l151–152). For test–retest reliability under identical raters and protocol, many readers expect ICC(2,1) or ICC(3,1) with explicit justification (two-way random vs mixed; absolute agreement vs consistency). Please clarify model assumptions and whether systematic bias between trials was considered clinically relevant.

Results

The results and tables are generally well presented and clear.

The use of <5th percentile as an impairment threshold is reasonable, but please justify the clinical reasoning for selecting the 5th and 2.5th percentiles.

The submaximal effort group showed lower z-scores (p11, l246–249), which is expected. However, stating that there were “no relevant differences” between groups (p11, l250–252) raises an interesting conceptual issue. Fear or pacing behavior itself may represent a meaningful PCC phenotype and deserves deeper interpretation.

Please reconcile the “good ICC” with the relatively wide limits of agreement. ICC can be inflated by large between-subject variability. Reporting SEM, MDC95, or coefficient of variation would strengthen clinical interpretability.

Discussion

The Discussion is generally well structured and demonstrates thoughtful interpretation. The absence of a learning effect is well addressed, and caution regarding single-item PEM measurement is appropriate.

However, several refinements would strengthen scientific depth:

You state that symptom severity may not limit short functional tasks (p18, l51–53). This requires more nuanced interpretation. Consider discussing:

(i) Pacing and anticipatory behavior decoupling symptoms from performance

(ii) The possibility that 30s duration may not trigger PEM in some individuals

(iii) Temporal mismatch between symptom measurement windows and test performance

Justification of Danish reference values based on similar inactivity prevalence (p20, l103–112) is reasonable but indirect. Consider framing this as approximate benchmarking rather than definitive impairment classification. Present raw repetition data as primary, with z-scores as contextual support.

Since anchor-based MID was not feasible, strongly consider emphasizing MDC95 as the measurement error threshold, with MID described as provisional.

Conclusions

The conclusion is concise and appropriately cautious regarding the need for future anchor-based MID validation (p21, l122–124).

Minor correction:

“suggest as minimal” → “suggest a minimal” (p21, l121–122).

Best wishes,

6. PLOS authors have the option to publish the peer review history of their article (what does this mean?). If published, this will include your full peer review and any attached files.). If published, this will include your full peer review and any attached files.

.

Reviewer #1: No

Reviewer #2: No

Reviewer #3: **Yes:** Yücel MakaracıYücel Makaracı

---

## [Author Response · Author response to Decision Letter 1]

31 Mar 2026

Our detailed point-to-point responses to the reviewers are uploaded as a separate document.

---

## [Decision Letter · Decision Letter 1]

14 Apr 2026

Measurement properties of the 30-second sit-to-stand test in post COVID-19 condition: results from the PYCNOVID randomised controlled trial

PONE-D-26-04488R1

Dear Dr. Radtke,

We’re pleased to inform you that your manuscript has been judged scientifically suitable for publication and will be formally accepted for publication once it meets all outstanding technical requirements.

An invoice will be generated when your article is formally accepted. Please note, if your institution has a publishing partnership with PLOS and your article meets the relevant criteria, all or part of your publication costs will be covered. Please make sure your user information is up-to-date by logging into Editorial Manager at Editorial Manager® and clicking the ‘Update My Information' link at the top of the page. For questions related to billing, please contact  and clicking the ‘Update My Information' link at the top of the page. For questions related to billing, please contact billing support..

Kind regards,

Mukhtiar Baig, Ph.D.

Academic Editor

PLOS One

Reviewers' comments:

Reviewer's Responses to Questions

**Comments to the Author**

1. If the authors have adequately addressed your comments raised in a previous round of review and you feel that this manuscript is now acceptable for publication, you may indicate that here to bypass the “Comments to the Author” section, enter your conflict of interest statement in the “Confidential to Editor” section, and submit your "Accept" recommendation.

Reviewer #3: All comments have been addressed

2. Is the manuscript technically sound, and do the data support the conclusions?

Reviewer #3: Yes

3. Has the statistical analysis been performed appropriately and rigorously? 

Reviewer #3: Yes

4. Have the authors made all data underlying the findings in their manuscript fully available?

Reviewer #3: Yes

5. Is the manuscript presented in an intelligible fashion and written in standard English?

Reviewer #3: Yes

6. Review Comments to the Author

Reviewer #3: Dear authors,

Thank you for revising the manuscript and for your responses. The revisions made have improved the clarity and overall structure of the manuscript.

7. PLOS authors have the option to publish the peer review history of their article (what does this mean?). If published, this will include your full peer review and any attached files.). If published, this will include your full peer review and any attached files.

.

Reviewer #3: No

---

## [Editor Report · Acceptance letter]

PONE-D-26-04488R1

PLOS One

Dear Dr. Radtke,

I'm pleased to inform you that your manuscript has been deemed suitable for publication in PLOS One. Congratulations! Your manuscript is now being handed over to our production team.

Kind regards,

on behalf of

Professor Mukhtiar Baig

Academic Editor

PLOS One